# ENTROPY ESTIMATES FOR GENERATIVE MODELS

**Kirill Struminsky, Michael Figurnov & Dmitry Vetrov**
Department of Computer Science
Higher School of Economics
Moscow, Russian Federation
`k.struminsky@gmail.com, michael@figurnov.ru, vetrovd@yandex.ru`

## ABSTRACT

Different approaches to generative modeling entail different approaches to evaluation. While some models admit test likelihood estimation, for others only proxy metrics for visual quality are being reported. In this paper, we propose a simple method to compute differential entropy of an arbitrary decoder-based generative model. Using this approach, we found that models with qualitatively different samples are distinguishable in terms of entropy. In particular, adversarially trained generative models typically have higher entropy than variational autoencoders. Additionally, we provide support for the application of entropy as a measure of sample diversity.

## 1 INTRODUCTION

In the past few years, an impressive variety of methods for generative modeling has been proposed, yet the approaches for model comparison seem to be at an early stage of development. Existing methods can sometimes be a source of misleading conclusions. For example, the inception score (Salimans et al. (2016)), may assign a high score to a model that gives as output $k$ samples representing each class in the dataset. In an effort to improve existing model comparison methods, we propose an approach based on differential entropy that captures sample diversity of the generative model.

In this work, we propose a method for entropy estimation for an arbitrary differentiable decoder model. We verify it in a setting where the differential entropy is known and then proceed with an empirical study of common generative models.

In our experiments, we observed an overall increase of entropy as the number of classes present in the training set increased, suggesting that differential entropy is indeed related to the coverage of various modes in the dataset. At the same time, we observed that the entropy of adversarially-trained generative models is always higher than the entropy of variational autoencoder. While variational autoencoders are known to have a high capacity to cover all of the modes in the training set, our results imply that this ability comes at cost of smaller sample variability.

## 2 ENTROPY ESTIMATION

Support of an arbitrary decoder-based generative model is by construction limited to a low-dimensional manifold embedded in a high dimensional space. As a result, to compute differential entropy $\mathcal{H} = -\mathbb{E}_{p(x)} \log p(x)$ of the model, one has to define the density of an arbitrary point of the manifold and then perform the integration. In the next section, we propose a method to estimate $\log p(x)$. Along with Monte-Carlo estimation of the expectation, the method gives a way to compute the differential entropy of a deep generative model.

### 2.1 DENSITY ESTIMATION IN AN AUXILIARY LATENT VARIABLE MODEL

Following Wu et al. (2016), we introduce a latent variable model to approximate the given generative model defined by decoder $d(\cdot)$. Standard Gaussian distribution $p(z) = \mathcal{N}(z|0, I)$ and isotropic Gaussian approximation $p(x|z, d) = \mathcal{N}(x|d(z), s^2 I)$ of decoder output induce marginal likelihood of observation $p(x|d)$.

We then adopt variational inference to estimate the log-marginal density $\log p(x|d)$. Suppose for a moment that the decoder is bijective and that for a given $x_0$ we know $z_0 = d^{-1}(x_0)$. In the limit of $s \to 0$, the posterior distribution is an improper distribution with all probability mass concentrated at the pre-image of $x_0$. With this in mind, we restrict the variational posterior to Gaussian distributions with known mean $z_0 = \mathbb{E}_{q(z)} z$.

To infer the covariance matrix of variational approximation $\Sigma$, we replace the likelihood term in the variational lower bound with a second-order Taylor approximation at $z_0$ with $\Delta z = z - z_0$. Optimization with respect to covariance matrix $\Sigma$ gives the following amortized variational approximation (see appendix A for details):

$$q(z|x,d) = \mathcal{N}\left(z|d^{-1}(x_0), \Sigma\right), \quad \Sigma = s^2 \left[ \left.\frac{\partial d}{\partial z}^T \frac{\partial d}{\partial z}\right|_{z=d^{-1}(x_0)} + s^2 I \right]^{-1} \tag{1}$$

Then variational approximation $q(z|x,d)$ can be used to compute a lower bound on marginal likelihood $p(x|d,s) \geq \mathbb{E}_q \log \frac{p(x,z|d,s)}{q(z|x,d)}$. Although the lower bound depends on the scale parameter $s$ and can be arbitrary high for sufficiently small $s$, in the next section we show that the lower bound can be decomposed into a sum of a scale-dependent term and a bounded decoder-specific term, which is related to density estimation on low-dimensional manifolds.

## 2.2 CONNECTION TO DENSITY ESTIMATION USING THE CHANGE OF VARIABLE FORMULA

As described in Gemici et al. (2016), a bijective differentiable transformation $d^{-1}$ of a random variable $z$ with density $p(\cdot)$ induces density $r(x) = p(d^{-1}(x)) \left| \frac{\partial d}{\partial z}^T \frac{\partial d}{\partial z} \right|_{z=d^{-1}(x)}^{-1/2}$. This formula is a generalization of a change of variable formula for probability distributions to situations where the image $x$ can lie in a higher-dimensional space.

We notice that for $x = d(z)$ the likelihood estimates from the previous section can be decomposed into two summands and a diminishing term (see appendix B for more details):

$$\mathbb{E}_q \frac{p(x,z|s)}{q(z|x)} = -\frac{D-d}{2} \log(2\pi s^2) + \left( \log p(d^{-1}(x)) - \frac{1}{2} \log \left| \frac{\partial d}{\partial z}^T \frac{\partial d}{\partial z} + s^2 I \right| \right) + o(s) \tag{2}$$

As $s$ tends to zero, the first term converges to $+\infty$, while the second term converges to the change of variable formula defined above.

## 2.3 ENTROPY ESTIMATION

To estimate model entropy we compute a Monte-Carlo estimate for a sufficiently small $s$. We found the regularization term $s^2 I$ to be crucial for stable computation of the log-determinant term across random model samples.

$$\mathcal{H} = \frac{1}{N} \sum_{z_i \sim p(z)} \left( \log p(z_i) - \frac{1}{2} \log \left| \left.\frac{\partial d}{\partial z}^T \frac{\partial d}{\partial z} + s^2 I\right|_{z=z_i} \right| \right). \tag{3}$$

# 3 EXPERIMENTS

## 3.1 ENTROPY OF A UNIFORM DISTRIBUTION ON A HALF-SPHERE

To verify the entropy estimate method, we consider a decoder $d(\cdot) : \mathbb{R}^2 \to S_-^2$ that maps a standard Gaussian distribution to a uniform distribution on a half-sphere. Differential entropy of a uniform distribution on a half-sphere is equal to $\log(2\pi) \approx 1.8379$.

Entropy estimates for various scale parameters and $N = 5 \cdot 10^5$ samples are reported in table 1. Additionally, a standard deviation of density estimate among samples is presented. By design of the experiment, each point has the same true density. In practice, we observe a decrease in standard deviation as the scale parameter $s$ tends to zero.

Table 1: Entropy estimates for a uniform distribution on a half-sphere. The estimate converges to the true entropy value as the scale parameter $s$ tends to zero.

| True | $s = 10^{-1}$ | $s = 10^{-2}$ | $s = 10^{-3}$ | $s = 10^{-4}$ |
|---|---|---|---|---|
| $\approx 1.8379$ | $1.9160 \pm 0.0032$ | $1.8428 \pm 0.0009$ | $1.8388 \pm 0.0003$ | $1.8379 \pm 0.0001$ |

## 3.2 COMPARISON OF DEEP GENERATIVE MODELS

We estimate entropy of four common generative models: VAE by Kingma & Welling (2013), GAN by Goodfellow et al. (2014), WGAN by Arjovsky et al. (2017), WGAN GP by Gulrajani et al. (2017). The four models were trained on MNIST and Fashion MNIST datasets for 64 epochs using an implementation from Lee (2017) with Gaussian input noise.

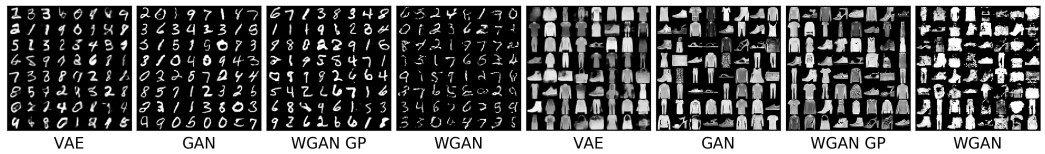

VAE  GAN  WGAN GP  WGAN  VAE  GAN  WGAN GP  WGAN

Figure 1: Generative model samples for MNIST and Fashion MNIST.

As shown in Figure. 1, the variational autoencoder samples tend to be blurrier, while WGAN samples were significantly noisier than the samples from other models. Entropy estimates reported for $s = 10^{-4}$ in table 2 reflect these specifics. Firstly, we observe that the variational autoencoder has notably lower entropy. This effect can be attributed to the posterior variance underestimation, which is specific for variational inference. Secondly, the high entropy of WGAN matches the high noisiness of model samples. Finally, WGAN GP and GAN have visually indistinguishable samples, but the slightly higher entropy of the former model is quantitative evidence for its higher sample diversity.

Table 2: Entropy estimates underline the difference between variational autoencoders and adversarially trained models.

| Dataset / Model | VAE | GAN | WGAN GP | WGAN |
|---|---|---|---|---|
| MNIST | -98.0 | -82.1 | -75.1 | -41.4 |
| Fashion MNIST | -117.2 | 2.9 | 6.7 | 25.4 |

## 3.3 SENSITIVITY TO CLASS EXCLUSION

We train a VAE on subsets of MNIST, containing all digits up to $k$-th digit. This experiment is intended to model situations in which a generative model misses important subsets in the dataset.

We then measure entropy of the resulting 10 generative models and report it for $s = 10^{-4}$ in table 3. While the size of the training set and diversity of training samples increase monotonically, in our experiments we did not observe a strictly monotonic increase in model entropy.

For MNIST we observe two entropy drops when images for digits "1" and "7" are added. These digits share similar shape and have, on average, significantly higher likelihood on test set than other digits. Therefore, they lower the overall entropy of the generative model.

Table 3: Entropy estimates for generative models trained on first $k$ classes of the dataset.

| $\leq k$ | $k = 0$ | $k = 1$ | $k = 2$ | $k = 3$ | $k = 4$ | $k = 5$ | $k = 6$ | $k = 7$ | $k = 8$ | $k = 9$ |
|---|---|---|---|---|---|---|---|---|---|---|
| MNIST | -177.7 | -191.0 | -146.7 | -130.9 | -129.9 | -107.3 | -104.4 | -110.5 | -104.7 | -105.7 |

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

## A   DERIVATION OF COVARIANCE ESTIMATE

Firstly, we replace the data-term in ELBO with a second-order Taylor approximation in $z_0 = z$:

$$\mathbb{E}_{q(z|x,\phi)} \log p(x|z, \theta) \tag{4}$$

$$\approx \left[ \log p(x|z_0, \theta) + \mathbb{E}_{q(z|x,\phi)} \left[ \frac{\partial \log p(x|z, \theta)}{\partial z} \Delta z + \frac{1}{2} \Delta z^T \frac{\partial^2 \log p(x|z, \theta)}{\partial z^2} \Delta z \right] \right. \tag{5}$$

$$\tag{6}$$

Both expectations in the approximation can be computed analytically and lead to the following approximation:

$$\mathbb{E}_{q(z|x,\phi)} \log p(x|z, \theta) \approx \log p(x|z_0, \theta) + \frac{1}{2} \operatorname{tr} \left[ \frac{\partial^2 \log p(x|z, \theta)}{\partial z^2} \Sigma \right] \tag{7}$$

$$\mathbb{E}_{q(z|x,\phi)} \log p(x|z, \theta) - \mathrm{KL}(q(z|x, \phi)||p(z)) \approx \tag{8}$$

$$\log p(x|z_0, \theta) + \frac{1}{2} \left[ \operatorname{tr} \left( \frac{\partial^2 \log p(x|z, \theta)}{\partial z^2} - I \right) \Sigma - z_0^T z_0 + d + \log |\Sigma| \right] \tag{9}$$

For a negative-definite matrix $\frac{\partial^2 \log p(x|z, \theta)}{\partial z^2}$ the latter approximation has a unique optimum point $\Sigma = (I - \frac{\partial^2 \log p(x|z, \theta)}{\partial z^2})^{-1}$. For the Gaussian decoder distribution $p(x|z, d) = \mathcal{N}(x|d(z), sI)$ and $x = d(z_0)$ the Hessian is negative-definite

$$\frac{\partial^2 \log p(x|z, \theta)}{\partial z^2} = - \left( \frac{1}{s^2} \frac{\partial d^T}{\partial z} \frac{\partial d}{\partial z} + \frac{\partial^2 d}{\partial z^2}(d(z_0) - x) \right) = -\frac{1}{s^2} \frac{\partial d^T}{\partial z} \frac{\partial d}{\partial z}, \tag{10}$$

and the optimal covariance for the ELBO approximation is

$$\Sigma = s^2 \left[ \left. \frac{\partial d^T}{\partial z} \frac{\partial d}{\partial z} \right|_{z=d^{-1}(x_0)} + s^2 I \right]^{-1}. \tag{11}$$

## B   ASYMPTOTIC REPRESENTATION OF ELBO

Again, we replace the quadratic term in likelihood $p(x|z, d)$ with a Taylor approximation:

$$\mathbb{E}_{q(z|x,d)} \log p(x|z, d) = -\frac{D}{2} \log(2\pi s^2) - \frac{1}{2s^2} \mathbb{E}_\epsilon ||x - d(z_0 + \Sigma^{1/2}\epsilon)||_2^2 \tag{12}$$

$$= -\frac{D}{2} \log(2\pi s^2) - \frac{1}{2s^2} \operatorname{tr} \left[ \frac{\partial^2 \log p(x|z, \theta)}{\partial z^2} \Sigma \right] + o(s) \tag{13}$$

$$= -\frac{D}{2} \log(2\pi s^2) - \frac{1}{2} \operatorname{tr}(I + s^2 \frac{\partial^2 \log p(x|z, \theta)}{\partial z^2})^{-1} + o(s) \tag{14}$$

$$= -\frac{D}{2} \log(2\pi s^2) - \frac{d}{2} + o(s). \tag{15}$$

The KL-term can be represented as follows:

$$\mathrm{KL}(q(z|x, d)||p(z)) = \frac{1}{2}(\operatorname{tr} \Sigma + z_0^T z_0 - d - \log |\Sigma|) \tag{16}$$

$$= \frac{1}{2}(z_0^T z_0 - d - d \log s^2 + \log \left| \frac{\partial d^T}{\partial z} \frac{\partial d}{\partial z} + s^2 I \right|) + o(s) \tag{17}$$

Subtracting KL-term from the average likelihood we get

$$\mathbb{E}_{q(z|x,d)} \frac{p(x,z|d)}{q(z|x,d)} = -\frac{D-d}{2} \log(2\pi s^2) + \left( \log p(d^{-1}(x)) - \frac{1}{2} \log \left| \frac{\partial d^T}{\partial z} \frac{\partial d}{\partial z} + s^2 I \right| \right). \quad (18)$$

## C   COMPARISON WITH VAE LOWER BOUNDS

Other variational approximations can be used to compute a lower bound for $\log p(x|d)$. For instance, one may fix a decoder and train a VAE encoder to estimate entropy $\mathcal{H}_d \leq -\mathbb{E}_q \log \frac{p(x,z)}{q(z|x)}$. However, in practice these estimates appear to be inexact.

We fit an encoder for GAN and WGAN GP models trained on MNIST dataset from subsection 3.2. We use an isotropic Gaussian decoder approximation $p(x|z,d) = \mathcal{N}(x|d(z), sI)$ with $s = 10^{-4}$ as a proxy for likelihood. In table 4 we report the entropy upper bounds along with the scale correction $\mathcal{H}_q = -\mathbb{E}_{p(x)} \mathbb{E}_q \log \frac{p(x,z|\theta)}{q(z|x,\phi)} - \frac{D-d}{2} \log(2\pi s^2)$.

Entropy upper bounds are reported for a basic encoder $q_A(z|x,\phi) = \mathcal{N}(z|\mu_\phi(x), \Sigma_\phi(x))$ and for a "decoder-aware" approximation $q_B(z|x,\phi) = \mathcal{N}(z|d^{-1}(x), \Sigma_\phi(x))$. In all four cases, amortized approximations for posterior parameters are too loose. In particular, a trivial estimate for the decoder mean $z_0 = d^{-1}(x)$ decreases the estimate by an order of magnitude. Additionally, due to gradient optimization issues, upper bounds with a full covariance matrix result in significantly weaker approximations. Thus amortized variational inference is inapplicable to entropy estimation.

Table 4: Entropy upper bounds using a VAE encoder versus the reported entropy

| Model | Reported entropy | $q_A$, diag. $\Sigma$ | $q_A$, full $\Sigma$ | $q_B$, diag. $\Sigma$ | $q_B$, full $\Sigma$ |
|---|---|---|---|---|---|
| GAN | -82 | 25595 | 25857 | 234 | 334 |
| WGAN GP | -75 | 25057 | 25471 | 198 | 302 |

