# OpenReview forum: "Entropy Estimates for Generative Models"
_ICLR.cc/2018/Workshop — Reject_

### Official Review · AnonReviewer2 · 2018-03-03
**Claim of GANs producing higher entropy is likely wrong**

**Rating:** 3
**Confidence:** 4

**Review:**

Summary:
This paper proposes an estimator for the differential entropy of decoder-based generative models as a means for evaluation. Using their estimator, the authors find that GANs and WGANs have higher entropy than a VAE when evaluated on MNIST and Fashion MNIST (which is likely due to a sign flip).

Review:
At a conceptual level, I wonder how useful differential entropy is as a metric for evaluating decoder-based generative models like GANs. Given that GANs are used to learn degenerate distributions (with undefined/negative infinite differential entropy), it is safe to assume that having large differential entropy is not a concern in this line of research.

Although I did not follow all the steps of the derivation, the estimator (Equation 3) seems to have a wrong sign, since:
– the entropy seems to decrease with increasing s and increasing gradient norms
– the extremely surprising finding that according to this estimator, GANs produce higher entropy estimates than VAEs
– based on a comparison with the ELBO representation in Equation 2 (which is also missing log)

What do you mean by "differential entropy of a uniform distribution on a half-sphere is equal to log(2pi)"? I'd expect the differential entropy of a degenerate distribution such as a distribution on a sphere to be undefined/negative infinity.

Similarly, in 2.2 you provide a formula for a density "where the image x can lie in a higher-dimensional space [than z]." Please clarify what you mean by density here, since degenerate distributions don't have probability densities. Are you referring to a more general notion such as Radon-Nikodym derivatives?

Minor:
"improper" -> "degenerate" in first paragraph on page 2

---

### Official Review · AnonReviewer3 · 2018-03-08
**I don't think this is accurate**

**Rating:** 4
**Confidence:** 4

**Review:**

The authors propose a method for estimating the entropy of a generative model from its samples. The resulting estimates do not seem to make sense. The authors claim that VAEs have lower entropy than GANs, which contradicts all earlier literature on this topic. My guess is that this is due to the Taylor approximation used to approximate the posterior of the latents in their method: this will work if the decoder is approximately one to one, which tends to be true for VAEs to a larger extent than for GANs.

---

### Official Review · AnonReviewer1 · 2018-03-17
**Not confident entropy can be accurately estimated in implicit models**

**Rating:** 5
**Confidence:** 5

**Review:**

Evaluating generative models, and especially implicit models where the likelihood is intractable, an outstanding problem. The authors propose using entropy as a way to measure sample diversity. Entropy requires calculating an expectation with repect to the distribution's likelihood: they lower bound this quantity using typical variational techniques, and in the case of implicit models, assume a Gaussian likelihood.

The statistical desiderata is very nice, but unfortunately, practical evaluation is complicated by computational considerations. Like Wu et al. (2016), I'm not confident where this method is applicable because it compounds onto the same problems as likelihood evaluation overall. Namely, GANs do not have such tractable likelihoods, and any evaluation metrics which rely on them should carefully note limitations in assumptions that do in fact assume they're tractable. The scale parameter assumption is verified only on a toy problem two-dimensional half-sphere.

It's also unclear how the variational lower bound plays a role: model comparison with lower bounds can be highly suspect. Is AIS applicable as in Wu et al. (2016) for tighter bounds?

---

### Public Comment · ~Oriol_Vinyals1 · 2018-02-17
**Please Fix Length**

Your paper violates by a few lines the 3 page limit (see https://iclr.cc/Conferences/2018/CallForWorkshops). Please send us a fixed version of your PDF at iclr2018.programchairs@gmail.com by the end of Monday, February 19th, or else we will reject your paper.

Thanks,
ICLR2018 Program Chairs

---

### Decision · Program_Chairs · 2018-03-20
**ICLR 2018 Workshop Acceptance Decision**

**Decision:**

Reject

**Comment:**

Based on the reviews, this paper has not been accepted for presentation at the ICLR workshop. However, the conversation and updates can continue to appear here on OpenReview.